# Near-Infrared In Vivo Imaging of Claudin-1 Expression by Orthotopically Implanted Patient-Derived Colonic Adenoma Organoids

**DOI:** 10.3390/diagnostics14030273

**Published:** 2024-01-26

**Authors:** Sangeeta Jaiswal, Fa Wang, Xiaoli Wu, Tse-Shao Chang, Ahmad Shirazi, Miki Lee, Michael K. Dame, Jason R. Spence, Thomas D. Wang

**Affiliations:** 1Division of Gastroenterology, Department of Internal Medicine, University of Michigan, Ann Arbor, MI 48109, USA; jaiswals@umich.edu (S.J.); wang1343@outlook.com (F.W.); wuxls@126.com (X.W.); leemiki@umich.edu (M.L.); mdame@med.umich.edu (M.K.D.); spencejr@med.umich.edu (J.R.S.); 2Department of Mechanical Engineering, University of Michigan, Ann Arbor, MI 48109, USA; tsechang@umich.edu; 3Division of Integrative System and Design, University of Michigan, Ann Arbor, MI 48109, USA; shirazi@umich.edu; 4Department of Biomedical Engineering, University of Michigan, Ann Arbor, MI 48109, USA

**Keywords:** colonoid, adenoma, imaging, peptide, confocal, endoscopy

## Abstract

Background: Claudin-1 becomes overexpressed during the transformation of normal colonic mucosa to colorectal cancer (CRC). Methods: Patient-derived organoids expressed clinically relevant target levels and genetic heterogeneity, and were established from human adenoma and normal colons. Colonoids were implanted orthotopically in the colon of immunocompromised mice. This pre-clinical model of CRC provides an intact microenvironment and representative vasculature. Colonoid growth was monitored using white light endoscopy. A peptide specific for claudin-1 was fluorescently labeled for intravenous administration. NIR fluorescence images were collected using endoscopy and endomicroscopy. Results: NIR fluorescence images collected using wide-field endoscopy showed a significantly greater target-to-background (T/B) ratio for adenoma versus normal (1.89 ± 0.35 and 1.26 ± 0.06) colonoids at 1 h post-injection. These results were confirmed by optical sections collected using endomicroscopy. Optical sections were collected in vivo with sub-cellular resolution in vertical and horizontal planes. Greater claudin-1 expression by individual epithelial cells in adenomatous versus normal crypts was visualized. A human-specific cytokeratin stain ex vivo verified the presence of human tissues implanted adjacent to normal mouse colonic mucosa. Conclusions: Increased claudin-1 expression was observed from adenoma versus normal colonoids in vivo using imaging with wide field endoscopy and endomicrosopy.

## 1. Introduction

Colorectal cancer (CRC) is a major contributor to the worldwide tumor burden and is a leading cause of cancer-related deaths in the US [1,2]. The adenoma-carcinoma sequence is regarded as the primary mechanism for CRC development [3]. Conventional colonoscopy uses white light illumination to detect and resect colonic adenomas. Widespread patient acceptance of this procedure has led to increased early detection of CRC and a dramatic reduction in incidence and mortality [4]. This in vivo imaging procedure is sensitive to grossly visible structural changes, such as polyps and masses. However, >25% of all adenomas may be missed and can result in interval cancers [5]. Moreover, flat and subtle appearing lesions can be easily missed [6]. Imaging methods based on molecular expression rather than structural changes may reduce the miss rates of pre-malignant lesions. Fluorescence provides high contrast images to localize adenomas for endoscopic resection. Endoscopes that are sensitive to fluorescently labeled molecular probes are being developed to target the detection of specific cancer biomarkers overexpressed by adenomas [7]. The NIR spectrum minimizes tissue autofluorescence, hemoglobin absorption, and tissue scattering.

Genetically engineered mice have been used extensively as a pre-clinical model to verify the in vivo performance of emerging imaging technologies [8]. For example, the *CPC*;*Apc* mouse was constructed to somatically delete an *APC* allele under Cre regulation. Adenomas develop spontaneously in the distal colon [9]. This model accurately reflects the genetic mechanism for CRC, as adenomatous polyposis coli (Apc) mutations are found in >80% of sporadic tumors [10]. However, clinically relevant genetic diversity and molecular heterogeneity are lacking. The properties are needed to rigorously evaluate targeted imaging strategies. Organoids are an emerging platform for pre-clinical use as a model to evaluate in vivo imaging strategies [11,12]. Organoids can be cryopreserved and expanded later for stable, long-term use. Colon organoids (colonoids) have been developed in culture from stem cells to mimic the biological properties of adenomas and normal colonic mucosa. Colonoids can be implanted orthotopically in the colon of immunosuppressed mice. This location provides an intact microenvironment with physiological vasculature to sustain colonoid viability [13]. Imaging instruments have been developed with sufficiently small dimensions to perform repetitive in vivo monitoring of colonoids implanted in mouse colons to visualize specific tumor uptake of molecular probes.

Claudin-1 is a membrane protein that forms tight junctions between epithelial cells that populate colonic crypts to form a barrier against paracellular transport [14,15]. Loss of polarity occurs during cancer transformation to provide space for the movement of tumor cells and delivery of cancer nutrients. The claudin-1 gene is found downstream in the Wnt signaling pathway and provides b-catenin-binding sites in the promoter region to activate transcription [16,17]. Wnt signaling is activated by a loss of APC proteins or by activation of b-catenin mutations [18,19]. The Wnt signaling pathway plays an important role in the development of sporadic CRC [20]. Also, claudin-1 overexpression has been shown to mediate the pathogenesis of inflammatory bowel disease (IBD) and colitis-associated CRC [21]. Pathological findings include impaired goblet cell differentiation, deferred epithelial recovery, sustained inflammation, and crypt hyperplasia. Moreover, claudin-1 has been found to be upregulated in sessile serrated adenomas (SSAs) [22]. These pre-malignant lesions can transform into CRC via *BRAF^V600E^* mutations. Detection of claudin-1 may be able to differentiate SSAs from hyperplastic polyps.

A peptide specific for claudin-1 was previously identified and validated [23]. Wide-field endoscopy was used to visualize uptake by adenomas in vivo. Also, claudin-1 expression in the distal colon of *CPC*;*Apc* mice was observed in vivo with sub-cellular resolution using a single axis confocal endomicroscope [24]. In the dual axes configuration, fluorescence is collected off-axis to minimize the capture of scattered light [25]. The rejection of unwanted light dramatically increases the dynamic range of detection [26]. This configuration allows for optical sections to be collected in vertical (perpendicular to the tissue surface) as well as horizontal (parallel to the tissue surface) planes. The vertical view provides the basilar-to-luminal direction and the natural progression of epithelial diseases. Evolving relationships among tissue microstructures can be appreciated, and the depth of biological phenomena relative to the mucosal surface can be accurately determined. Here, we aim to demonstrate the use of patient-derived colonoids implanted orthotopically in mouse colons as a pre-clinical model to visualize claudin-1 expression in vivo. An NIR fluorescently labeled peptide specific for claudin-1 was administered intravenously. The presence of adenoma colonoids were localized using wide-field endoscopy, and claudin-1 expression by epithelial cells was visualized with sub-cellular resolution using dual axes confocal endomicroscopy.

## 2. Materials and Methods

### 2.1. Peptide Specific for Claudin-1

For imaging using the wide-field endoscope, RTSPSSR, a 7 amino acid peptide specific for claudin-1, hereafter RTS*, was synthesized using a PS3 automatic synthesizer (Protein Technologies Inc., Tucson, AZ, USA), and labeled with Cy5.5, a near infrared red (NIR) fluorophore (cyanine5.5 NHS ester, Lumiprobe 27020, Hunt Valley, MD, USA) at the C-terminus using a GGGSK linker, hereafter RTS*-Cy5.5 [7]. SPTSSRR (scrambled control), hereafter SPT*, was also labeled with Cy5.5, hereafter SPT*-Cy5.5. The peptides were purified to >99% by HPLC and were lyophilized for storage at −80 °C.

For imaging using the dual axes confocal endomicroscope, the peptides were labeled with IRDye800CW via a GGGSC linker to maximize tissue depth (Biomatik, Wilmington, DE, USA). The unlabeled RTS* peptide was mixed with 2 mg of peptide and 1 mg of IRDye800CW maleimide (LI-COR Biosciences, Lincoln, NE, USA) in 2 mL of coupling buffer (0.1 M sodium phosphate, 0.5 mM TCEP, pH 7.4). The reaction was performed in N_2_ for 2 h at room temperature (RT). The IRDye800CW-labeled peptides were purified using reversed-phase high-performance liquid chromatography (RP-HPLC). Crude peptides were characterized using mass spectrometry and evaluated by HPLC for purity >99%. The peptides were lyophilized for storage at −80 °C.

### 2.2. Patient-Derived Colonoids in Culture

#### 2.2.1. Colonoid Origins

Patient-derived colonoids were previously established in culture from adenoma and normal human colonic mucosa [27,28,29]. The colonoids were provided by the Translational Tissue Modeling Laboratory (TTML; https://www.umichttml.org (accessed on 23 January 2024), University of Michigan IRB REP00000105 with unregulated designation). A total of 12 adenoma and 5 normal specimens were screened for claudin-1 expression. Based on immunofluorescence staining, 2 adenoma colonoids with high claudin-1 expression and 1 normal colonoid with low claudin-1 were selected. The adenoma colonoids were derived from tissue biopsies of tubular adenomas (#584, 20 mm; 61 years, male and #590, 35 mm; 58 years, female). The normal colonoid was derived from a deceased donor (#87; 21 years, male). The genomic profiles of these colonoids (publicly available at E-MTAB-9339 and E-MTAB-4698) were published previously [27,29].

#### 2.2.2. Colonoid Cultures

Cultures were grown in Matrigel (Corning, Thermo Fisher Scientific, Glendale, AZ, USA, #354234) diluted to 8 mg/mL with growth media in 6-well tissue culture plates (USA Scientific CytoOne, Ocala, FL, USA, #CC7682-7506). Cultures were passaged by triturating and dissociating Matrigel in cold DPBS while centrifuging at 300× *g*. Plating was performed using 2.5 µM CHIR99021 (Tocris (Minneapolis, MN, USA), #4423) and 10 µM Y27632 (Tocris, #125410).

Normal colonoids were cultured in a human colonoid medium (HCM) containing 50% L-WRN conditioned medium (source of Wnt3a, R-spondin-3 and Noggin) [30], advanced DMEM/F-12 (Gibco, #12634028), N-2 media supplement (Gibco, Thermo Fisher, Grand Island, NY, USA), #17502048), B-27 supplement minus vitamin A (Gibco, #12587010), 1 mM N-Acetyl-L-cysteine (Sigma-Aldrich (Burlington, MA, USA), #A9165), 2 mM GlutaMax (Gibco, #35050-061), 10 mM HEPES (Gibco, #15630080), 50 units/mL penicillin, 0.05 mg/mL streptomycin (Gibco, #15070063), 100 µg/mL Primocin (InvivoGen (San Diego, CA, USA); #ant-pm-1), 100 ng EGF/mL (R&D Systems Inc. (Minnneapolis, MN, USA), #236-EG), 10 µM SB202190 (Sigma-Aldrich (Burlington, MA, USA), #S7067), 500 nM A83-01 (Tocris, #293910), and 10 µM Y27632 (Tocris, #125410).

The adenoma organoid (#590) was cultured in Stemline Complete medium containing StemlineTM Keratinocyte Medium II (Sigma, #S0196), Stemline Growth Supplement (Sigma, #S9945), 2 mM GlutaMax, 4 mM L-glutamine, and 50 µg/mL Primocin. Prior to harvesting for implantation, the colonoid cultures were treated with 5 µM Y27632 for 18 h. The adenoma organoid (#584) was cultured in 50% of the above Stemline complete medium and 50% of the above HCM.

#### 2.2.3. Colonoid Preparation for Implantation

Cultures were harvested from Matrigel in cold DPBS, triturated 30× with a 1 mL pipette tip, and centrifuged at 300× *g* for 3 min at 4 °C. The colonoid pellet was resuspended in 10 mL cold DPBS and mechanically disassociated with the gentleMACS Octo Dissociator (Miltenyi Biotec (Bergisch Gladbach, Germany), #130-096-427) using the programs h_Tumor_01.01 followed by m_Lung-01.01. The colonoid fragments were further dissociated by 20× pipetting using a 1 mL pipette tip. Large fragments were removed over a 100 µm BSA-coated cell strainer (Corning, #DL 352360). A slow centrifugation at 100× *g* was done to reduce single cell content. The cell aggregates were resuspended in cold DPBS supplemented with 5% Matrigel, and 10 µM Y27632. ~2.5–5 × 10^5^ cell aggregates in 200 µL were implanted per mouse [31]. All plasticware, including gentleMACS C-tubes (Miltenyi, #130-093-237), were treated with 0.1% BSA in DPBS to reduce colonoid adherence.

#### 2.2.4. Pre-Clinical Colonoid Model

All animal experiments were performed with approval from the Institutional Animal Care and Use Committee (IACUC) at the University of Michigan. Mice were housed 2–5/cage in a specific pathogen-free facility, given a standard chow diet and water ad libitum, and exposed to a 12 h light/12 h dark circle. Human-derived colonoids were implanted in the colons of 8–10 week old female NOD/SCID (Jackson laboratory (Bar Harbor, ME USA), 005557, NOD Cg-Prkdc < scid > ll2rg < tm1Wjl > SzJ) mice. Before implantation, mice were provided with 2.5–3% dextran sulfate salt (DSS) water for 5 days. Anesthesia was induced and maintained via a nose cone with inhaled isoflurane mixed with oxygen at concentrations of 2–4% and a flow rate of 0.5 L/min. Approximately 2.5–5 × 10^5^ cell aggregates in 200 μL were injected intrarectally. Following injection, the rectum was closed using tissue adhesive (Santa Cruz Biotechnology (Dallas, TX, USA), #SC361931) to promote retention of engrafted colonoids. A total of *n* = 8 mice were implanted with adenoma colonoids (2 with #584, 6 with #590), and *n* = 7 mice were implanted with normal colonoids (#87). Colonoid growth in mouse colons was monitored weekly.

### 2.3. Claudin-1 Expression by Patient-Derived Colonoids

#### 2.3.1. Preparation of Paraffin-Embedded Specimens

The cultured colonoids were fixed in 4% paraformaldehyde (PFA) at 4 °C overnight and embedded in a histogel. The fixed colonoids were then processed and embedded in paraffin. The paraffin-embedded colonoid blocks were sectioned with 5 μm thickness. After completion of imaging, the mouse colon was resected and transected at 1/3 of the distance from the anus. The colon was divided longitudinally and rinsed with cold PBS. The tissues were fixed in 10% neutral buffered formalin for 48 h before transfer into 70% ethanol and then embedding in paraffin. Tissue sections were cut at 5 μm thickness for routine histology (H & E).

#### 2.3.2. Immunohistochemistry (IHC)

Colonoid and tissue sections were processed for IHC staining. Deparaffinization and antigen retrieval were performed on unstained sections prior to staining as described previously [7]. Sections were immersed in 3% H_2_O_2_ for 30 min, blocked with normal serum for 30 min, and incubated overnight with anti-claudin-1 antibody (Abcam, Boston, MA, USA, #ab15098) at 4 °C. Antigen binding was detected with a VECTASTAIN Elite ABC HRP Kit containing biotinylated rabbit IgG (Vectorlabs (Mowry Ave Newark, CA, USA), #PK6101) and chromogen 3,3′-diaminobenzidine (DAB; Sigma #D3939) per the manufacturer’s instructions. Slides were counterstained with Harris hematoxylin. The digital images were collected using a standard brightfield/epifluorescence microscope (Zeiss (New York, NY, USA), #Axioskop 2 plus).

### 2.4. Claudin-1 Expression in Patient Specimens

FFPE blocks of human colon specimens were obtained from the Department of Pathology at the University of Michigan (IRB HUM00038437). Colonoids and mouse colon tissue sections were prepared as described above. Sections were deparaffinized, and antigen retrieval was performed by boiling slides in a sodium citrate buffer. Blocking was performed by incubating sections with 10% normal serum for 1 h at RT. Sections were first stained by incubating with 5 μM of RTS*-Cy5.5 for 5 min followed by washing 3× with PBS. Next, antibody staining was performed by incubating with anti-claudin-1 antibody (Abcam, #ab15098) at 4 °C. Goat anti-rabbit IgG antibody conjugated with AF488 (Life Technologies (Carlsbad, CA ,USA), #A-11029) was used as a secondary antibody and incubated with the tissue sections for 2 h at RT. For human cytokeratin staining, sections were incubated with anti-cytokeratin (CAM 5.2, BD Biosciences, Franklin Lakes, NJ, USA, #345779) at RT for 1 h followed by incubation with goat anti-mouse IgG conjugated with AF488 (Thermo Fisher Scientific, #28175) for 2 h at RT. Fluorescence images were collected using an inverted confocal microscope (Leica (Deerfield, IL, USA), #SP5). The average fluorescence intensity was measured using custom Matlab R2022a (Mathworks) software.

### 2.5. Wide-Field Imaging of Implanted Colonoids

A rigid, wide-field endoscope (Karl Storz Veterinary Endoscopy-America, Goleta, CA, USA) with white light illumination was inserted into the distal mouse colon. Prior to imaging, the mice were fasted for 4–6 h. If viable colonoids were found, landmarks were recorded using (1) the distance between the endoscope tip and the anus and (2) the clockwise location of the colonoids. The fluorescently labeled peptides were then intravenously administered at a concentration of 200 µM diluted in 200 µL of PBS. This instrument was adapted for excitation of Cy5.5 at l_ex_ = 671 nm [32]. Images were collected at 1-h post-injection, which corresponded to the time frame for peak uptake [24]. After 3 days, the control peptide was administered systemically in the same group of mice. Images were collected in vivo using the previously identified landmarks at the same time point. Images were processed and analyzed using custom Matlab (Mathworks) software. The relative intensities within the regions of interest (ROI) were calculated by taking the ratio of the fluorescence and corresponding reflectance images [33]. A total of 3 ROIs with dimensions of 20 × 20 μm^2^ were picked at random from the colonoid (target) and adjacent normal colonic mucosa (background). The average fluorescence intensity from each ROI was used to calculate the target-to-background (T/B) ratio.

### 2.6. Confocal Imaging of Implanted Colonoids

A side-view dual axes confocal endomicroscope was used to collect optical sections in vivo from the implanted adenoma and normal colonoids to visualize claudin-1 expression with sub-cellular resolution [25]. Lateral and axial resolutions of 1.75 and 7.5 mm, respectively, were achieved using excitation of IRDye800CW at l_ex_ = 785 nm. The instrument had an outer diameter of 4.2 mm and could be inserted repetitively into the mouse colon without introducing trauma. Vertical cross-sectional images (perpendicular to tissue surface) were collected with dimensions of 900 × 310 mm^2^. Horizontal cross-sectional images (parallel to the tissue surface) were collected with dimensions of 660 × 660 mm^2^. NIR fluorescence images were collected in vivo a 1 h following intravenous injection of RTS*-IRDye800CW (200 µM in 200 µL PBS). Images were processed using custom MATLAB (Mathworks) software.

### 2.7. Toxicity Analysis

Normal human colon cells (CCD 841 CoN) were cultured in RPMI containing 10% fetal bovine serum (FBS) and a 1% antibiotic solution (penicillin and streptomycin). Approximately 10^4^ cells were seeded in a 96 well plate and incubated overnight at 37 °C in the presence of 5% CO_2_. The cells were washed with 1× PBS and treated with RTS*-Cy5.5 and RTS*-IRDye800. After 24 h post treatment, 10 μL of MTT (5 mg/mL) was added to the media and incubated at 37 °C. After 2 h, the media was carefully removed, and dimethyl sulfoxide was added. The absorbance at λ = 570 nm was measured using a microtiter plate reader.

### 2.8. Statistics

A one-way ANOVA model with terms for 9 conditions was used to evaluate peptide cytotoxicity. Fluorescence intensities from confocal and in vivo imaging were quantified using custom MATLAB (Mathworks) software. Fluorescence intensities were log transformed to improve normality. Comparisons between two cohorts were evaluated using either paired or unpaired *t*-tests, as described in the text. GraphPad Prism was used to determine statistical significance.

## 3. Results

### 3.1. Peptide Specific for Claudin-1

RTSPSSR was previously identified using phage display methods [23]. This sequence was scrambled as SPTSSRR for use as a control. The sequences were labeled with Cy5.5 using a GGGSK linker to characterize specific binding to claudin-1 expressed by patient-derived colonoids (Appendix A). Peak fluorescence emission for RTS*-Cy5.5 and SPT*-Cy5.5 were found in the NIR spectrum between ~650–900 nm (Appendix A). RTS* was labeled with IRDye800 via a GGGSC linker for in vivo imaging to minimize tissue autofluorescence, hemoglobin absorption, and tissue scattering (Appendix A). Peak absorbance and emission for RTS*-IRDye800 were found in the NIR spectrum between ~600–950 nm (Appendix A). The fluorescently labeled peptides were purified to >95% by HPLC. Mass spectrometry analysis was performed for RTS*-Cy5.5 and SPT*-Cy5.5 (Appendix A) and for RTS*-IRDye800 (Appendix A). The mass-to-charge (*m*/*z)* ratios measured were in agreement with expected values. RTS*-Cy5.5 and RTS*-ITDye800 did not show any toxicity to normal cells (Appendix A).

### 3.2. Patient-Derived Colonoids in Culture

A brightfield (BF) image is shown of patient-derived adenoma colonoids (arrows) in culture in Figure 1a. Histology (H & E) of colonoid sections shows the adenoma phenotype in Figure 1b. An IHC stain confirmed strong claudin-1 expression by adenoma colonoids (Figure 1c). Brightfield, histology, and IHC are shown for the normal colonoids and confirmed reduced expression of claudin-1 in Figure 1d–f.

### 3.3. Claudin-1 Expression by Patient-Derived Colonoids

Immunofluorescence (IF) was performed to assess staining of adenoma and normal colonoid sections ex vivo. Strong fluorescence intensity was seen with anti-claudin-1-AF488 and RTS*-Cy5.5 to the surface (arrows) of epithelial cells located within dysplastic crypts of adenoma colonoids (Figure 2a,b). The merged images in Figure 2c show co-localization of antibody and peptide binding. A Pearson’s correlation coefficient of r = 0.74 was measured. A reduced signal was observed for antibody and peptide binding to the epithelial surface (arrows) of normal colonoids (Figure 2d,e). A correlation r = 0.66 was measured from the merged image (Figure 2f). The fluorescence intensities were quantified, and the mean values measured for anti-claudin-1-AF488 and RTS*-Cy5.5 were found to be significantly greater for adenomas versus normal (Figure 2g,h, respectively).

### 3.4. Claudin-1 Expression in Patient Specimens

Two adenoma colonies with distinct molecular expression patterns and one normal colonoid were chosen for in vivo implantation in mouse colons. Tissue sections from the respective patients were obtained to evaluate claudin-1 expression levels for comparison with that for cultured colonoids. IF using anti-claudin-1-AF488 and RTS-Cy5.5 showed strong staining to surface (arrows) of epithelial cells in dysplastic crypts for the first adenoma (Figure 3a). IHC was performed in an adjacent section and confirmed strong claudin-1 expression. Histology (H & E) confirmed the adenoma phenotype. Similar results were found for the second adenoma (Figure 3b). IF using anti-claudin-1-AF488 and RTS-Cy5.5 showed reduced staining to the surface (arrows) of epithelial cells in normal crypts (Figure 3c). IHC from an adjacent section confirmed baseline claudin-1 expression. Histology (H & E) confirmed the normal phenotype.

### 3.5. Wide-Field Imaging of Implanted Colonoids

Patient-derived adenoma and normal colonoids were implanted orthotopically in the colons of immunocompromised mice. The presence of viable colonoids was monitored using wide-field endoscopy with white light illumination (Figure 4a). Claudin-1 expression by colonoids was evaluated in vivo by injecting RTS*-Cy5.5 via the tail vein. NIR fluorescence images were collected at 1 h post-injection. A strong signal was seen from the adenoma colonoids (arrow) (Figure 4b). NIR fluorescence images were then collected from the same region in the same animals after injection of SPT*-Cy5.5 (scrambled control) 3 days later using previously identified landmarks. A minimal signal was detected (Figure 4c). The Minimal signal was seen from the implanted normal colonoids with both the target (RTS*-Cy5.5) and control (SPT*-Cy5.5) peptides (Figure 4d–f). The NIR fluorescence intensities were quantified, and the mean T/B ratio for RTS*-Cy5.5 was significantly greater than that for SPT*-Cy5.5 (Figure 4g). Also, the mean T/B ratio for RTS*-Cy5.5 was significantly greater from adenomas versus normal colonoids (Figure 4h).

### 3.6. Confocal Imaging of Implanted Colonoids

Claudin-1 expression was visualized in vivo using a side view dual axes confocal endomicroscope. NIR fluorescence images from adenoma and normal colonoids were collected in vertical (XZ) cross-sections at 1 h post-injection of RTS*-IRDye800 (300 mM, 200 µL PBS), as shown in Figure 5a,b (Appendix A). Mucosal structures, including crypts (arrow), lumens (l), and lamina propria (lp), could be seen. Images with dimensions of 900 × 310 mm^2^ were collected in the vertical plane. NIR fluorescence images were also collected in horizontal (XY) cross-sections at a depth of Z = 165 µm, as shown in Figure 5c,d (Appendix A). Individual epithelial cells from dysplastic crypts in adenoma organoids were visualized with strong peptide staining. Baseline claudin-1 expression by epithelial cells in normal crypts was observed.

### 3.7. Ex Vivo Validation of Implanted Colonoids

After imaging was completed, the animals were euthanized, and the colon was resected. IF staining of tissue sections was performed ex vivo to further validate claudin-1 expression by the implanted colonoids. Human-specific anti-cytokeratin (hCKT) was used to distinguish human from mouse tissues. The implanted patient-derived adenoma colonoid showed strong fluoresence intensity from hCKT-AF488 to identify human (Hu) tissues (Figure 6a). By comparison, adjacent mouse (Mu) colonic mucosa showed minimal intensity. Immunostaining with anti-claudin-1-AF561 and RTS*Cy5.5 confirmed strong staining to human (Hu) by comparison with mouse (Mu) tissues. Histology (H & E) showed the presence of patient-derived adenoma colonoids implanted adjacent to normal mouse colon. The implanted human (Hu) normal colonoids showed intense fluorescence staining with hCKT-AF488 versus adjacent mouse colons (Figure 6b). Immunostaining with anti-claudin-1-AF561 and RTS*Cy5.5 showed minimal intensity for normal human (Hu) and mouse (Mu) tissues. Histology (H & E) showed the presence of patient-derived normal colonoids implanted adjacent to normal mouse colons.

## 4. Discussion

### 4.1. Pre-Clinical Model of CRC

Here, we demonstrated a pre-clinical model of CRC using patient-derived colonic organoids implanted orthotopically in mouse colons. This model was used to evaluate the specific uptake of a fluorescently labeled peptide specific for claudin-1 with in vivo imaging. Human colonoids provide clinically relevant levels of target expression, and the orthotopic location generates representative vasculature to evaluate ligand delivery. Co-staining of the peptide and anti-claudin-1 antibody was observed with high correlation to colonoids in culture and to the patient-derived tissues used to establish the colonoids. Wide-field fluorescence images were collected from the colons of immunocompromised mice at 1 h post-injection of the peptide. The mean intensity measured from adenomas was significantly greater than that from normal tissues, and the average value from the target peptide was significantly greater than that from the scrambled control. Peptide binding to crypts in the epithelium was verified using vertical and horizontal cross-sectional images collected with a side-view dual axes confocal endomicroscope. A human-specific cytokeratin stain confirmed the presence of patient-derived tissues within normal mouse colonic mucosa ex vivo. These results demonstrate promise for implanted organoids to serve as a pre-clinical model to evaluate fluorescently labeled ligands for specific binding to adenomas prior to a more complex clinical study.

### 4.2. Patient-Derived Colonoids

Optical sections collected in vivo over time can be used to monitor the function of claudin-1, including the formation of tight junctions, maintenance of cell polarity, and regulation of paracellular transport [14]. The mechanism for target upregulation during the cancer transformation process may be elucidated. Claudin-1 was found to be overexpressed in the original colonic adenoma from the patient specimens. Overexpression was preserved in the patient-derived colonoids and in xenografts implanted orthotopically in mouse colons. Orthotopic transplantation of colonoids from culture can also be performed to study therapeutic response in vivo. The human organoids used in this study were derived from normal and transformed patient tissues that were cultured in vitro to recapitulate the tumor development process. We have previously developed efficient orthotopic transplantation methods to enable physiological tumor growth in pre-clinical cancer models [34] and are now using this technique to advance the clinical translation of novel imaging methodologies. Human genomic expression has been authenticated, and genetic variations have been characterized. Therefore, a pre-clinical model with orthotopically implanted human colonoids is a promising approach to evaluate the in vivo uptake of a peptide specific for claudin-1 for the early detection of CRC.

### 4.3. Claudin-1-Targeting Peptide

Claudin-1 was identified as a promising target for early detection of CRC based on a 2.5 fold increase in gene expression in human adenomas versus normal colonic mucosa. Adenomas are pre-malignant; thus, claudin-1 serves as an early target [35]. The sequence RTS* was selected using phage display methods by biopanning against the purified extracellular loop mimetic target [23]. This sequence was found to be specific for claudin-1 using siRNA knockdown, competition, and co-localization assays. RTS* was found to have a binding affinity of 42 nmol and an onset of 1.2 min. This sequence demonstrated specific uptake in vivo by adenomas that develop spontaneously in a genetically engineered *CPC*;*Apc* mouse using topical and intravenous administration. Serum stability was measured with a half-life of T_1/2_ = 1.9 h. Specific peptide binding to adenomas was confirmed with in vivo optical sections collected in horizontal cross-sections at a depth of 100 mm using a single axis confocal endomicroscope [24]. This peptide does not show toxicity to different organs after intravenous injection [24]. Previously, an 18 amino acid peptide was derived from the intracellular and first transmembrane domain of claudin-1 to inhibit entry of HCV at a post-binding step [36]. This observation demonstrated potential for a new class of inhibitors to target the entry process for viruses. Also, a nanoparticle was demonstrated that specifically targets increased claudin-1 expression to reduce the blood–brain barrier integrity [37]. This finding can be used to develop longitudinal monitoring of tight junction protein expression during aging.

### 4.4. In Vivo Imaging of Implanted Colonoids

In vivo optical sections were collected in vertical cross-sections (XZ) using a dual axes confocal endomicroscope [25]. This view of the epithelium allowed for expression of cancer biomarkers to be visualized with a depth >310 mm, which was achieved using optics that collected light off-axis to generate a high dynamic range. A large image FOV > 900 mm was achieved using post-objective scanning with a miniature, microsystems mirror. Images from adenoma and normal colonic mucosa showed mucosal features with a strong signal over the depth of the epithelium. Dysplastic crypts appeared with variable sizes, uneven spacing, and disorganized structures. Normal crypts appeared with relatively uniform size, even spacing, and as tall columnar structures. NIR fluorescence images were collected in the horizontal (XY) plane at a depth of Z = 165 mm and showed the presence of individual epithelial cells. The instrument diameter of 4.2 mm allowed for repetitive use in the mouse colon without introducing trauma. Tumor uptake and clearance of the peptide can be characterized in vivo over time. The same mice can be used to evaluate both the target and control peptides to achieve statistical rigor using a minimum number of animals. Previously, confocal images were collected in horizontal (parallel to tissue surface) cross-sections only where the biology was fairly similar across the field of view (FOV) [24].

### 4.5. Application of Patient-Derived Colonoid Transplant Model

Human colonoids provide clinically relevant target expression levels that are representative of the patient population. Tumors were implanted in the orthotopic location to generate relevant vasculature to characterize ligand delivery. Also, this pre-clinical model offered an intact microenvironment to evaluate in vivo peptide uptake. The target peptide sequence was scrambled to provide a rigorous control to confirm specific ligand binding to claudin-1. Greater fluorescence intensities were collected from adenomas using the target versus control peptide. This observation suggests specific binding to the intended target rather than an enhanced permeability and retention (EPR) effect [38]. The adenomas appeared in vivo with different morphologies, including flat, small, and grossly visible. Many of these lesions were barely distinguishable with white light illumination alone. These results support the potential for future clinical translation of this targeted imaging methodology to identify flat and subtle pre-malignant lesions that can be easily missed using conventional white light colonoscopy. This model shows that gene expression is recapitulated in cultured and implanted organoids. In addition, orthotopic implantation provides an intact tumor microenvironment. Therefore, in addition to validation of imaging probes, this model can also be used to test the efficacy of therapeutic agents.

## 5. Conclusions

This study demonstrates a patient-derived orthotopic transplant model for pre-clinical validation of an NIR fluorescently labeled peptide using in vivo imaging. Several conclusions can be drawn:Patient-derived adenoma and normal colonoids implanted orthotopically in colonic mucosa of immunocompromised mice can be imaged.Implanted colonoids have a similar morphology to flat and subtle human colonic adenomas.Fluorescently labeled peptides can distinguish overexpressed cell surface targets, such as claudin-1, expressed by adenoma versus normal colonoids in vivo.Multi-model imaging with wide-field endoscopy and dual axes confocal endomicroscopy could be used to localize and optically section colonoids, respectively.In summary, a human pre-clinical model has demonstrated in vivo uptake of an NIR-labeled peptide specific for claudin-1 to localize pre-malignant colonic lesions in vivo using wide-field endoscopy and confocal endomicroscopy by generating high fluorescence contrast.

## Figures and Tables

**Figure 1 diagnostics-14-00273-f001:**
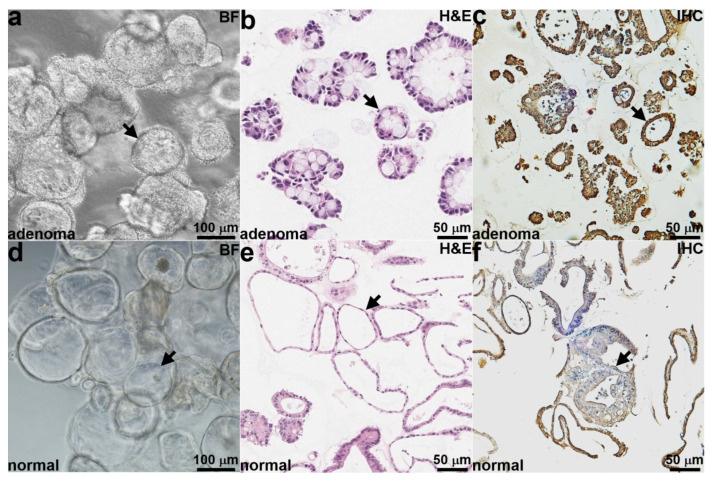
Patient-derived colonoids in culture. (**a**) Brightfield (BF) and (**b**) histology (H & E) images are shown of patient-derived adenoma colonoids (arrows). (**c**) IHC shows claudin-1 expression by adenoma colonoids (arrow). (**d**) Brightfield, (**e**) histology (H & E), and (**f**) IHC images are shown of patient-derived normal colonoids (arrows).

**Figure 2 diagnostics-14-00273-f002:**
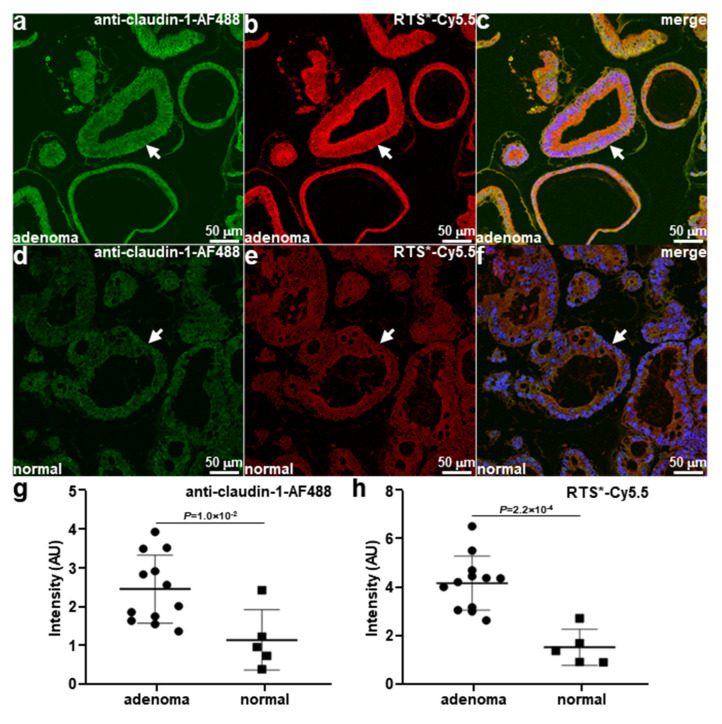
Claudin-1 expression by patient-derived colonoids. Adenoma colonoids (arrows) were stained with (**a**) anti-claudin-1-AF488 and (**b**) RTS*-Cy5.5. (**c**) Merged image shows the co-localization of antibody and peptide with a Pearson’s correlation coefficient of r = 0.74. Patient-derived normal colonoids (arrows) were stained with (**d**) anti-claudin-1-AF488 and (**e**) RTS*-Cy5.5. (**f**) Merged image shows a correlation of r = 0.66. The mean fluorescence intensity from adenoma (*n* = 12) was significantly greater than that from normal (*n* = 5) colonoids for (**g**) anti-claudin1-AF488 and (**h**) RTS*-Cy5.5 with *p* = 1.0 × 10^−2^ and 2.2 × 10^−4^, respectively, by two-way *t*-test.

**Figure 3 diagnostics-14-00273-f003:**
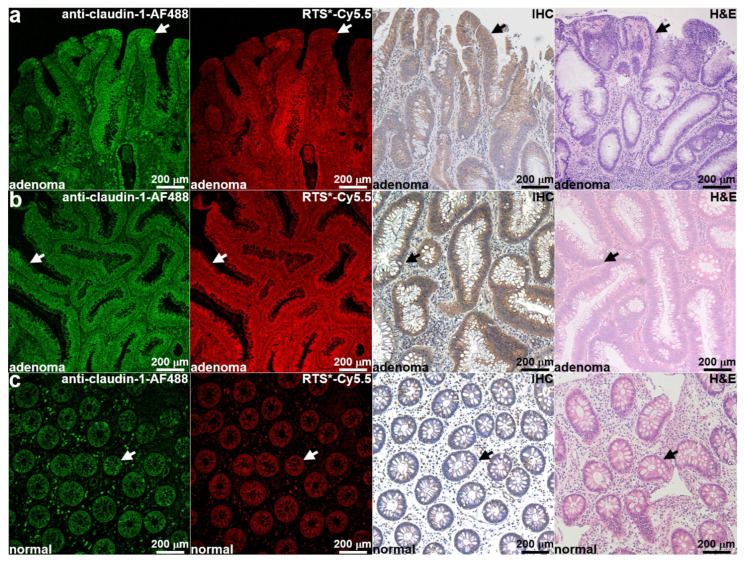
Claudin-1 expression by human colon specimens. (**a**) Immunofluorescence with anti-claudin-1-AF488 and RTS*-Cy5.5 for the first adenoma (#584) used to develop patient-derived colonoids showed strong staining at the surface (arrow) of epithelial cells in dysplastic crypts. IHC confirmed increased claudin-1 expression in this adenoma. Histology (H & E) of adenoma is shown. (**b**) Immunostaining with anti-claudin-1-AF488 and RTS*-Cy5.5, IHC, and histology (H & E) showed similar results for the second adenoma (#590). (**c**) Immunofluorescence with anti-claudin-1-AF488 and RTS*-Cy5.5 from the normal (#87) colon showed reduced staining for claudin-1. IHC and histology (H & E) confirm these results.

**Figure 4 diagnostics-14-00273-f004:**
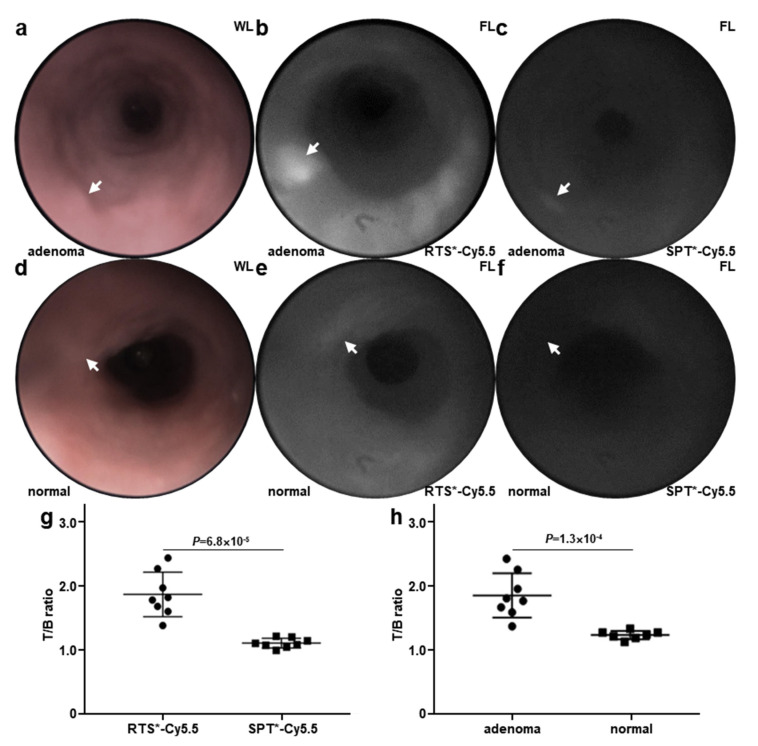
Wide-field colonoid images. (**a**) White light (WL) image is shown of patient-derived adenoma colonoid (arrow) implanted in the colon of an NOD/SCID mouse. (**b**) NIR fluorescence (FL) image shows increased uptake of RTS*-Cy5.5 by the adenoma colonoid (arrow). (**c**) Fluorescence image of SPT*-Cy5.5 (control) in the same animal collected 3 days later shows minimal signal from the adenoma colonoid (arrow). (**d**) White light image is shown of patient-derived normal colonoid (arrow). (**e**) NIR fluorescence image shows reduced uptake of RTS*-Cy5.5. (**f**) Fluorescence image of SPT*-Cy5.5 (control) is shown for the same animal. (**g**) Mean (±SD) T/B ratio for RTS*-Cy5.5 (*n* = 8) is significantly greater than that for SPT*-Cy5.5 (*n* = 8) at 1 h post-injection, 1.89 ± 0.35 versus 1.12 ± 0.07, respectively, *p =* 6.8 × 10^−5^. (**h**) Mean (±SD) T/B ratio for RTS*-Cy5.5 (*n* = 8) in human adenoma is significantly greater than that for normal colonoids (*n* = 7) at 1 h post-injection, 1.89 ± 0.35 and 1.26 ± 0.06, respectively, *p =* 1.3 × 10^−4^. Paired *t*-tests were performed on log-transformed data.

**Figure 5 diagnostics-14-00273-f005:**
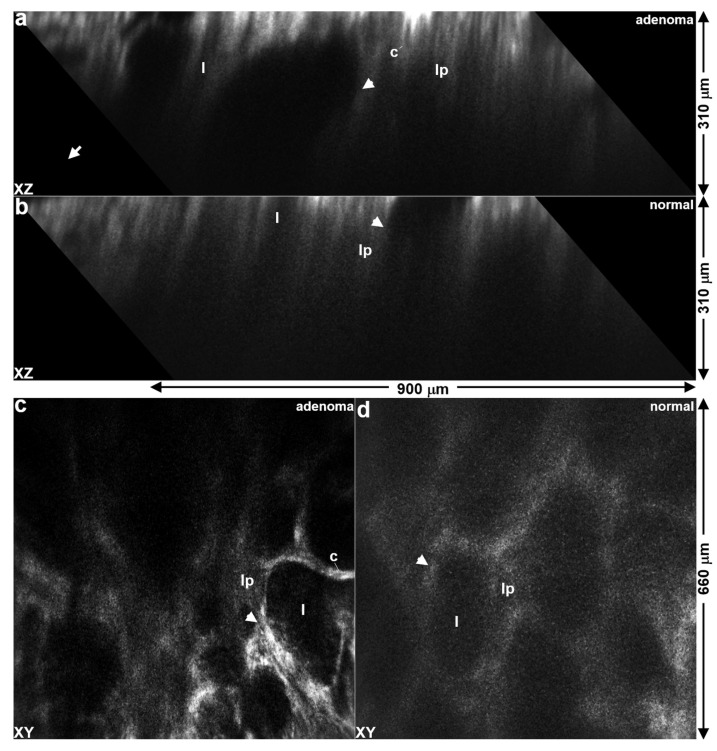
In vivo optical sections of claudin-1 expression. Vertical cross-sectional images with dimensions of 900 × 310 mm^2^ are shown for (**a**) adenoma and (**b**) normal colonoids implanted in mouse colons. Horizontal cross-sectional images are shown with dimensions of 660 × 660 mm^2^ of (**c**) adenoma and (**d**) normal colonoids. NIR fluorescence image was collected at ~1 h following intravenous injection of RTS*-IRDye800 (200 µM in 200 µL PBS). Key: crypt (arrow), c—colonocytes, l—lumen, lp—lamina propria.

**Figure 6 diagnostics-14-00273-f006:**
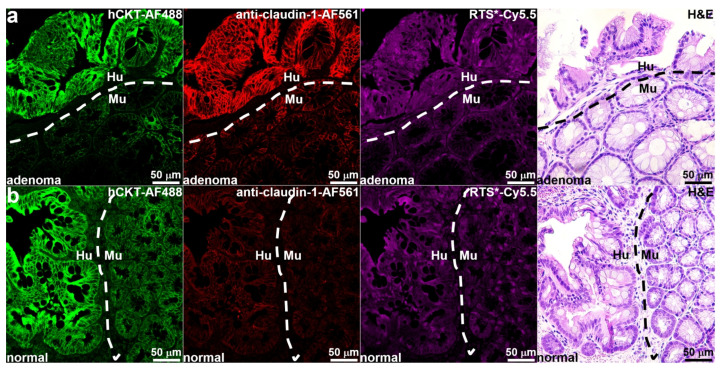
Ex vivo validation of implanted patient-derived colonoids. (**a**) IF stain using AF488-labeled anti-human cytokeratin (hCKT) shows strong presence of human (Hu) adenoma adjacent to mouse (Mu) colons from tissues obtained after completion of imaging. IF stain using anti-claudin-1-AF561 and RTS*-Cy5.5 confirmed strong expression of claudin-1 by adenoma. Histology (H & E) of human (Hu) implanted next to mouse (Mu) tissues is shown. (**b**) IF stain using anti-hCKT-AF488 shows presence of human (Hu) normal tissues adjacent to mouse (Mu). IF stain using anti-claudin-1-AF561 and RTS*-Cy5.5 confirmed minimal expression of claudin-1 by normal tissues. Histology (H & E) of normal human (Hu) implanted next to mouse (Mu) tissues is shown.

## Data Availability

The reported data, including the Appendix A, are available from the corresponding authors upon request.

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
