# Peer review of "Near-Infrared In Vivo Imaging of Claudin-1 Expression by Orthotopically Implanted Patient-Derived Colonic Adenoma Organoids"

_diagnostics, 2024, doi:10.3390/diagnostics14030273_

Round 1

Reviewer 1 Report

Comments and Suggestions for Authors

This manuscript demonstrated a strategy for localizing pre-malignant colonic lesions in vivo by using NIR-labeled peptide ligand. These NIR-labeled target peptides ligands specific bind to Claudin-1 which is a tight junction protein that becomes overexpressed in colorectal cancer. Greater fluorescence intensity was collected from adenomas using the target  peptides ligands than the target peptide sequence was scrambled and in normal tissue. The advantages of this Caudin-1 specific fluorescent peptide labeling strategy were studied, but there are still some shortcomings and lack of discussion. It is suggested to accept this article after a major revision.

1.The RTS*-Cy5.5 and SPT*-Cy5.5 were used in the imaging of implanted colonoids. The structural, mass spectral and spectral characteristics of RTS*-Cy5.5 and SPT*-Cy5.5 need to be provided.

2.These NIR-labeled target peptide ligands are used for in vivo imaging. Their toxicity to normal cells needs to be discussed.

3.Pretreatment of target cells with ligands generally inhibits specific recognition of targeted probes. What effect does pretreatment of colorectal cancer cells with RTS* peptide have on the imaging of fluorescent-labeled peptides?  

Comments on the Quality of English Language

minor editing.

Author Response

Near-Infrared In Vivo Imaging of Claudin-1 Expression by Orthotopically Implanted Patient-Derived Colonic Adenoma Organoids

Reviewer 1

This manuscript demonstrated a strategy for localizing pre-malignant colonic lesions in vivo by using NIR-labeled peptide ligand. These NIR-labeled target peptides ligands specific bind to Claudin-1 which is a tight junction protein that becomes overexpressed in colorectal cancer.  Greater fluorescence intensity was collected from adenomas using the target peptides ligands than the target peptide sequence was scrambled and in normal tissue.  The advantages of this Caudin-1 specific fluorescent peptide labeling strategy were studied, but there are still some shortcomings and lack of discussion. I t is suggested to accept this article after a major revision.

1.The RTS*-Cy5.5 and SPT*-Cy5.5 were used in the imaging of implanted colonoids. The structural, mass spectral and spectral characteristics of RTS*-Cy5.5 and SPT*-Cy5.5 need to be provided.

Response:  We have included the mass spectra for RTS*-Cy5.5 and SPT*-Cy5.5 in Fig. S1 and the structural and spectral characteristics in Fig. S2.

2.These NIR-labeled target peptide ligands are used for in vivo imaging.  Their toxicity to normal cells needs to be discussed.

Response:  We have added results of an MTT assay for the NIR-labeled target that showed no peptide toxicity to normal cells in Fig. S3.

3.Pretreatment of target cells with ligands generally inhibits specific recognition of targeted probes. What effect does pretreatment of colorectal cancer cells with RTS* peptide have on the imaging of fluorescent-labeled peptides?

Response:  We have found previously that pretreatment of target cells with peptide ligands inhibits specific binding.  These results were shown previously in a competition assay by adding unlabeled RTS* to RTS*-Cy5.5 for incubation with human SW680 cells in vitro, and have been published (Fig. 4A, Ref 23).

Reviewer 2 Report

Comments and Suggestions for Authors

Manuscript Title “Near-Infrared In Vivo Imaging of Claudin-1 Expression by Orthotopically Implanted Patient-Derived Colonic Adenoma Organoids”

General comment:

The quality of this manuscript is OK, but there still are some tiny incomplete parts that need to revise as listed below,

Specific comment:

1.      Abstract; read like part of the introduction and no solid quantified data at all. A good abstract should include definition of keywords, experimental setup in short, result and essential discussions as well.

2.      Introduction; the description is OK

3.      Materials and methods; statements are clearly described. Either section or sub-section head is well defined and easy understanding

Sec 2.7, better elaborate the process of data analysis.

4.      Results, try to add some tables with solid data to illustrate the derived performance. There is none table listed in this manuscript.

5.      Discussion, suggest to extend the length of discussion, the author can shift some part in introduction to here to balance the ratio, otherwise too long introduction or too short discussion are weird to imply the own research finding, further the discussion is also recommended to categorized into several sub-section with specific section head to emphasize the topic of viewpoint.

6.       Conclusion, Too short the conclusion to make a strong ending

Author Response

Near-Infrared In Vivo Imaging of Claudin-1 Expression by Orthotopically Implanted Patient-Derived Colonic Adenoma Organoids

Reviewer 2

General comment:

The quality of this manuscript is OK, but there still are some tiny incomplete parts that need to revise as listed below,

Specific comment:

  1. Abstract; read like part of the introduction and no solid quantified data at all. A good abstract should include definition of keywords, experimental setup in short, result and essential discussions as well.

Response:  We have revised the Abstract to include quantified data, definition of keywords, description of experimental setup, results, and essential discussion.

  1. Introduction; the description is OK

Response:  We thank the reviewer for the kind remark.

  1. Materials and methods; statements are clearly described. Either section or sub-section head is well defined and easy understanding Sec 2.7, better elaborate the process of data analysis.

Response:  We have added headings for sections and sub-sections in Materials and Methods.  We have provided more details in section 2.7 to better describe the process used for data analysis.

  1. Results, try to add some tables with solid data to illustrate the derived performance. There is none table listed in this manuscript.

Response:  Our results were well described by images and plots, and we were not able to find use for tables.

  1. Discussion, suggest to extend the length of discussion, the author can shift some part in introduction to here to balance the ratio, otherwise too long introduction or too short discussion are weird to imply the own research finding, further the discussion is also recommended to categorized into several sub-section with specific section head to emphasize the topic of viewpoint.

Response:  We have categorized the Discussion into several sub-sections with specific section head to emphasize the topic of viewpoint.

  1. Conclusion, Too short the conclusion to make a strong ending

Response:  We have added more text to the Conclusion for a strong ending.

Reviewer 3 Report

Comments and Suggestions for Authors

Jaiswal et al. presented a Cy5.5-labeled peptide that was specific for Claudine-1 and used it for imaging colonoids that were derived from human patients and implanted orthotopically in mice. Compared to normal colonoids, adenoma colonoids showed significant higher uptake of the NIR peptide. This imaging probe presents new ways for diagnosing malignant adenoma from benign polyps, which is usually hard to differentiate using wide-field endoscopy. I recommend this manuscript to be published in its current form.

Author Response

Reviewer 3

Comments and Suggestions for Authors

Jaiswal et al. presented a Cy5.5-labeled peptide that was specific for Claudine-1 and used it for imaging colonoids that were derived from human patients and implanted orthotopically in mice. Compared to normal colonoids, adenoma colonoids showed significant higher uptake of the NIR peptide. This imaging probe presents new ways for diagnosing malignant adenoma from benign polyps, which is usually hard to differentiate using wide-field endoscopy. I recommend this manuscript to be published in its current form.

Response:  We thank the reviewer for the kind remarks.

Round 2

Reviewer 1 Report

Comments and Suggestions for Authors

The authors have addressed my concerns and the manuscript can be accepted now.  

Comments on the Quality of English Language

minor editing.